# T cell receptor convergence is an indicator of antigen-specific T cell response in cancer immunotherapies

Mingyao Pan, Bo Li*

Lyda Hill Department of Bioinformatics, The University of Texas Southwestern Medical Center, Dallas, United States

**Abstract** T cells are potent at eliminating pathogens and playing a crucial role in the adaptive immune response. T cell receptor (TCR) convergence describes T cells that share identical TCRs with the same amino acid sequences but have different DNA sequences due to codon degeneracy. We conducted a systematic investigation of TCR convergence using single-cell immune profiling and bulk TCRβ-sequence (TCR-seq) data obtained from both mouse and human samples and uncovered a strong link between antigen-specificity and convergence. This association was stronger than T cell expansion, a putative indicator of antigen-specific T cells. By using flow-sorted tetramer+ single T cell data, we discovered that convergent T cells were enriched for a neoantigen-specific CD8+ effector phenotype in the tumor microenvironment. Moreover, TCR convergence demonstrated better prediction accuracy for immunotherapy response than the existing TCR repertoire indexes. In conclusion, convergent T cells are likely to be antigen-specific and might be a novel prognostic biomarker for anti-cancer immunotherapy.

## Editor's evaluation

In this valuable and important study, the authors use cancer immunology datasets to study and discover a new biomarker for immune checkpoint blockade response. Not only does this work have the potential to be clinically impactful, but it also provides a deeper understanding of basic biology that can be applied to many different disease settings, and is supported by solid evidence.

*For correspondence:
bo.li@utsouthwestern.edu

Competing interest: The authors declare that no competing interests exist.

## Introduction

T lymphocytes, or T cells, are one of the most important components of the adaptive immune system. T cell receptors (TCRs) are protein complexes found on the surface of T cells that can specifically recognize antigens (*Marrack and Kappler, 1987*; *Davis and Bjorkman, 1988*). Through the combinational somatic rearrangement of multiple variable (V), diversity (D) (only for β chains), joining (J), and constant (C) gene segments, theoretically, the diversity of unique TCR α and β chains pairs can reach to $2 \times 10^{19}$ (*Pai and Satpathy, 2021*). Having such a wide variety of TCRs allows the recognition of numerous endogenous and exogenous antigens. However, due to codon degeneracy, which means multiple codons can encode the same amino acid (AA; *Trainor et al., 1984*), different V(D)J rearrangements can end up encoding the same TCR proteins, and this phenomenon is called TCR convergence (*Looney et al., 2019*). TCR convergence can be observed universally in almost every individual, but very few studies have been conducted to explain its biological significance.

While little is known about the functional impact of TCR convergence, codon degeneracy has been widely studied (*Gonzalez et al., 2019*; *McClellan, 2000*). The fact that the number of three-nucleotides codons (64) exceeds the number of encoding AAs (20) provides the basis of codon

degeneracy. The distribution of degeneracy among the 20 encoding AAs is uneven. Arginine, leucine, and serine are among the AAs with six corresponding codons, while others have one to four codons (*Crick et al., 1961*), and the origin of codon degeneracy remains the subject of debate. Some believe that codon degeneracy is the result of co-evolution in which codon assignment occurred by organisms inheriting parts of the codon set from precursor AAs (*Wong, 1975*). Some have tried to explain it through stereochemical interaction, where stereochemical specificity enables codons to selectively bind to assigned AAs (*Copley et al., 2005*). There are also other theories pertaining to this topic (*Crick et al., 1961*; *Di Giulio, 2004*; *Koonin and Novozhilov, 2009*). Apart from its elusive origins, the biological significance of codon degeneracy also varies in different scenarios. In the context of T cell immunity, the T cell repertoire not only results from somatic recombination but also is shaped by multiple selective pressures (*Jameson and Bevan, 1998*). Therefore, it is possible that TCR convergence is more than the mere consequence of codon degeneracy. It may provide unique insights into the antigen-driven TCR selection process.

T cells can be activated by their cognate antigens through the binding of the TCR to the peptide/ MHC complex (*Smith-Garvin et al., 2009*). Hence, antigen-specific T cells are crucial parts of the T cell 'army' and play dominant roles in eliminating pathogens or tumor cells. There is an increasing number of immunotherapies based on antigen-specific T cells, such as immune checkpoint blockade (ICB) or neoantigen vaccine therapies (*Postow et al., 2015a*; *Peng et al., 2019*), which further emphasize the necessity to study T cell antigen specificity. However, the enormous diversity of epitopes targeted by T cells and the highly polymorphic nature of MHC make it extremely challenging to identify the specificity of given T cells (*Newell and Davis, 2014*). Several studies have attempted to predict antigen-specific T cell responses by utilizing TCR characteristics such as diversity, clonality, or evenness scores. While there were cases where these signals were associated with the patient's survival (*Riaz et al., 2017*; *Tumeh et al., 2014*), in most cases these signals did not correlate with the outcomes (*Johnson et al., 2016*; *Robert et al., 2014*; *Amaria et al., 2018*; *Kidman et al., 2020*). This could be attributed to the weak correlation between these parameters and the antigen selection process. Here, we speculate that TCR convergence is a better indicator of antigen-driven selection. This is because, compared with a non-convergent TCR, a convergent TCR consumes more rearrangement resources, and the principle of parsimony (*Stewart, 1993*) implies a greater impact on the adaptive immune response for these convergent TCRs.

TCRs with similar CDR3 sequences shared similar binding structures and antigen specificity (*Dash et al., 2017*). This idea has led to several TCR similarity-based clustering algorithms, such as ALICE (*Pogorelyy et al., 2019*), TCRdist (*Dash et al., 2017*), GLIPH2 (*Huang et al., 2020*), iSMART (*Zhang et al., 2020*), and GIANA (*Zhang et al., 2021*), for studying antigen-driven T cell expansion during viral infection or tumorigenesis. While TCR convergence seems like a special case of similarity-based TCR clustering, it is conceptually different. By default, around 90% of the resulting TCR clusters by most previous algorithms are antigen-specific, while the ratio of some, like GLIPH2, is substantially lower (around 35%) (*Zhang et al., 2021*). Nevertheless, the fact that T cells within each convergent cluster share identical CDR3β AA sequence more likely represents shared antigen specificity compared to those with mismatches (*Dash et al., 2017*). The exploit of multiple nucleotide combinations for the same antigen specificity can potentially be a functional redundancy of adaptive immunity and reveal novel TCR selection process.

Over the last few years, high-throughput bulk TCRβ-sequencing (bulk TCR-seq) and single-cell immune profiling have become powerful tools for unraveling a variety of biological problems (*Liu et al., 2022*; *Zheng et al., 2017*). Previously generated datasets are excellent resources for re-analysis. Utilizing 3 single-cell immune profiling and 11 bulk TCR-seq datasets, we conducted a comprehensive analysis of TCR convergence. We began by investigating the basic features of convergent TCRs, including their degeneracies, publicity, and AA usage. Next, we explored the correlation between convergent and antigen-specific TCRs from different perspectives. Significant overlaps were found between convergent T cells and antigen-specific T cells across independent datasets. We proceeded to examine the phenotypes of convergent T cells in both mouse and human samples. As a result, convergent T cells exhibited gene signatures associated with cytotoxicity, memory, and exhaustion, related to antigen-specific T cells. Finally, we found that TCR convergence predicts improved clinical outcomes for cancer patients who received ICB therapies. Our work presents a new angle to study T cell repertoire and deepens our understanding of antigen-specific T cell selection.

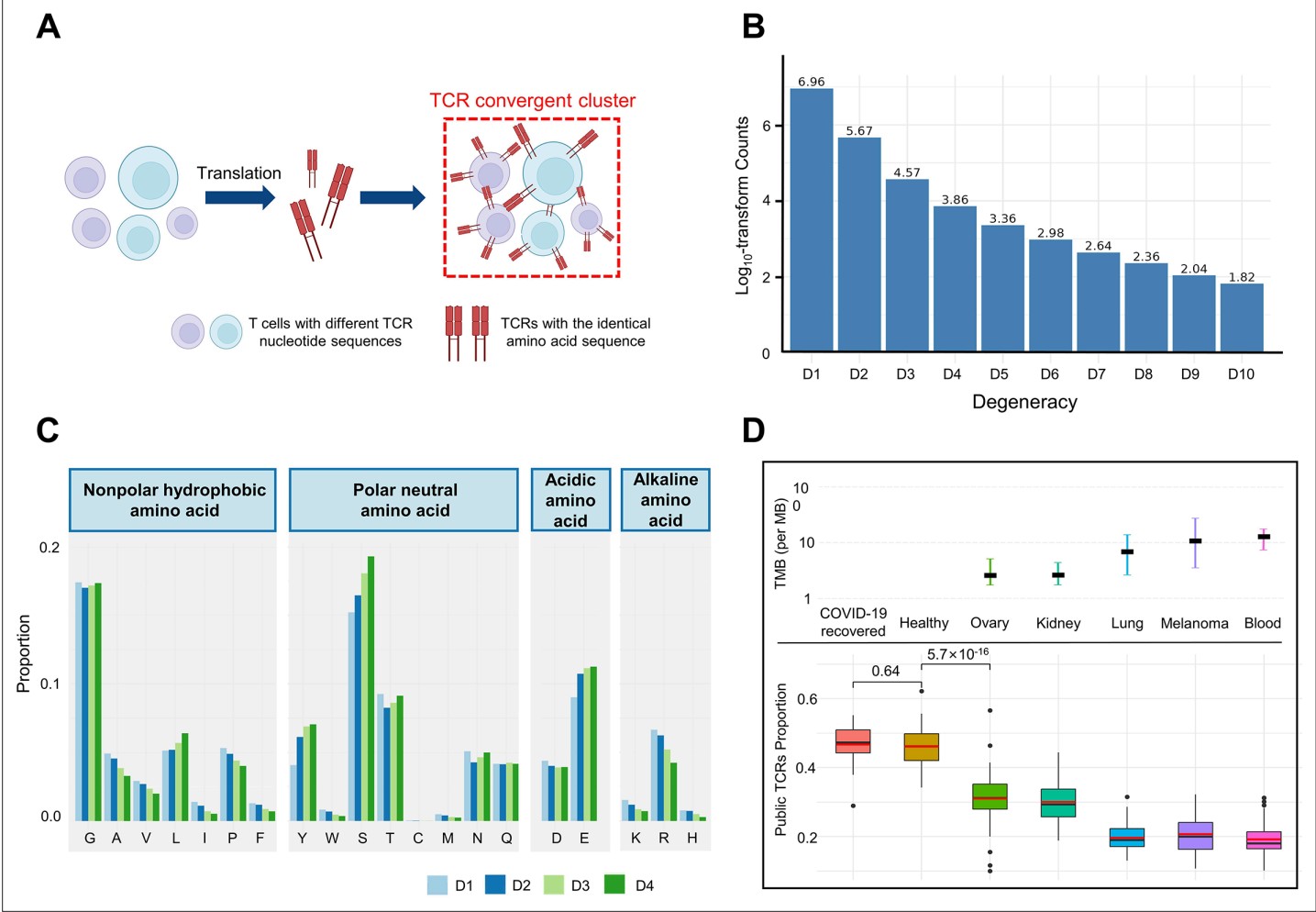

**Figure 1.** T cell receptor (TCR) convergence is different from publicity. (**A**) Cartoon illustration of TCR convergence, created with BioRender.com. T cells with different DNA sequences encode the same TCR due to codon degeneracy. (**B**) Log10-transformed counts of TCRs with different degrees of degeneracies. The x-axis indicates the number of distinct DNA sequences corresponding to the same TCR, or, the 'degree of degeneracy'. 'D1' refers to the sequences without any degeneracy. (**C**) The proportion of 20 amino acids in TCRs with degeneracies from one to four. Each of the 20 natural amino acids is represented on the x-axis by a one-letter abbreviation. (**D**) Tumor mutation burden (TMB) and the proportion of public TCRs in convergent TCRs among different cancer types. The upper panel shows the TMB of ovary, kidney, lung, melanoma, and blood cancer types. Cancer types are ordered from the lowest median TMB (left) to the highest median TMB (right). Lower panel shows the ratio of public TCRs among convergent TCRs in the respective cancer types. The sample size of each group: recovered COVID-19 patients (n=38), healthy control (n=50), ovary (n=45), kidney (n=19), lung (n=50), melanomas (n=29), blood (n=53). The statistical significance was calculated by Welch's t-test.

The online version of this article includes the following figure supplement(s) for figure 1:

**Figure supplement 1.** The length and variable gene usage of convergent T cell receptors (TCRs).

## Results

### TCR convergence is different from publicity

We defined the convergent TCRs as above described (*Figure 1A*) and analyzed their basic characteristics, including the degeneracies of the TCRs, sequence lengths, variable gene usage, and AA components using a bulk TCR dataset containing 666 blood samples from healthy donors (*Emerson et al., 2017*). Before analysis, quality control (QC) was performed to exclude CDR3 reads that did not meet the following two standards: (1) between 8 and 23 AAs in length; and (2) starting with cysteine and ending with phenylalanine. We randomly selected 50 samples, with a total of 9,567,751 that passing QC unique TCR used in the following analysis.

We defined the degeneracy of a TCR protein as the number of distinct clonotypes (defined by DNA sequences) encoding that TCR within one sample. In other words, a degeneracy equal to 1 means

non-convergent TCRs, while greater than 1 indicates convergent TCRs. As expected, convergent TCRs only comprised a small proportion (5.40%) of the total T cell population (*Figure 1B*). As TCR degeneracy increased from two to four, the number of detected TCRs decreased approximately 10-fold with each unit increase in degeneracy. Most convergent TCRs had a degeneracy of two (90.65%) or three (7.16%), while larger degeneracies were rare.

We studied the length distribution of the convergent TCR CDR3 regions, which peaked at 11 AAs (*Figure 1—figure supplement 1A*). While TCRs with 14–15 AA made up the greatest percentage of all the unique TCRs (*Figure 1—figure supplement 1B*), the proportion of convergent TCRs continually decreased after reaching 11 AA or longer. This result suggested that shorter TCR sequences appeared to be more conducive to TCR convergence. In terms of the correlation between TCR degeneracy and CDR3 lengths, CDR3s of 9 AA long displayed the greatest degeneracy on average (~2.5) (*Figure 1—figure supplement 1C*). From 11 AAs onward, the longer the sequences, the smaller the degeneracy. After reaching 17 AAs, the degeneracy remained constant. This is an unexpected observation, as the probability of the occurrence of codon degeneracy should increase with longer sequence lengths. We also analyzed the variable gene usage of convergent TCRs and observed that the usage of every TCRβ chain variable gene superfamily was relatively even (*Figure 1—figure supplement 1D*). TCR degeneracy was not significantly impacted by variable gene usage either (*Figure 1—figure supplement 1E*).

We further interrogated the average percentage of a given AA in TCRs with degeneracy from one to four. The first and last three AAs in the CDR3 sequences were not included since they are determined by the types of V or J genes. The result showed that, generally, convergent TCR favors polar neutral AAs and acidic AAs, particularly tyrosine (Y), serine (S), and glutamic acid (E), with alkaline AAs and nonpolar hydrophobic AA less favorable (*Figure 1C*). Arginine (R), leucine (L), and serine (S), each of which has six corresponding codons, have the largest codon degeneracy. However, leucine and arginine did not show the same level of relevance to TCR convergence as serine. While the ratio of leucine slightly increased in convergent TCR, the percentage of arginine even decreased as the TCR degeneracy increased. Tyrosine and glutamic acid, on the other hand, have only two codons each but were found in a higher percentage of convergent TCRs than non-convergent TCRs. The possibility of each above-mentioned AA (L, Y, S, E, R) in each degree of TCR degeneracy has been validated by two-sided binomial exact tests with p-values less than $2.2 \times 10^{-16}$ (except for the difference between 'D2' and 'D1' for leucine, whose p-value is $7.018 \times 10^{-7}$). These results indicate that the level of codon degeneracy is not a determinant of TCR degeneracy. Instead, the physiochemical properties of a given AA might have a greater impact on TCR convergence. Therefore, even though TCR convergence is the result of codon degeneracy, it exhibits an independent distribution and may carry a biological significance that is distinct from codon degeneracy.

Public TCRs are generated from VDJ recombination biases and are shared across different individuals, which might target common antigens, such as viral epitopes (*Emerson et al., 2017*). In contrast, TCRs targeting cancer neoantigens are mostly 'private' that are unique to each individual (*Madi et al., 2014*). Healthy individuals are expected to be exposed to common pathogens, which might induce public T cell responses. On the other hand, cancer patients have more neoantigens due to the accumulative mutation, which drives their antigen-specific T cells to recognize these 'private' antigens. This reduces the proportion of public TCRs in antigen-specific TCRs. A higher tumor mutation burden (TMB) would indicate a higher abundance of neoantigens, resulting in a lower ratio of public TCRs. Convergent recombination was claimed to be the mechanistic basis for public TCR response in many previous studies (*Quigley et al., 2010*; *Venturi et al., 2006*). As TCR convergence and publicity are conceptually similar, we next investigated the differences between the two by comparing the fractions of public TCRs within the convergent TCRs from healthy donors and patients with different cancers (*Figure 1D*). As expected, we observed a high overlap (47.92%) between the two in healthy donors as well as in another independent cohort consisting of patients who recovered from COVID-19 for more than 6 weeks (47.44%). Whereas for the cancer patients, the overlap between TCR convergence and publicity decreases for cancer types with higher TMB, which was estimated from previous studies (*Yarchoan et al., 2019*). The fraction of public TCRs within the convergence sequences dropped from 34.11% for the low mutation burden ovarian cancer to 21.34% for the highly mutated blood cancer samples. As the public TCRs in these cohorts are consistently defined by the Emerson 2017 cohort (*Emerson et al., 2017*), this difference is unlikely caused by unknown batch effects. Together, our

results indicated that TCR convergence and publicity represent two different biological processes that diverge in cancer patients by tumor mutation load.

## Convergent TCRs are more likely to be antigen-specific

To explore the relationship between convergent T cells and antigen-specific T cells, we began by examining datasets with known TCR antigen specificity. To be precise, since all the T cells are specific to some antigen(s) during positive thymic selection, the term 'antigen-specific' means T cells with an ongoing or memory antigen-specific immune response in this context. In 2020, 10× Genomics detected the antigen specificity of T cells using highly multiplexed peptide-MHC multimers (*10xGenomics, 2020*) from peripheral blood mononuclear cells (PBMCs) samples of four healthy individuals. We picked out all convergent TCRs within each donor and examined their overlaps with antigen-specific multimer[+] TCRs. Interestingly, while convergent TCRs only consisted of a small proportion of all unique clones, they were dominantly enriched for antigen-specific TCRs (*Figure 2A*). In these four donors, only 12.91, 33.32, 71.65, and 6.24% of the unique TCRs were multimer positive, respectively. However, 88.00, 88.28, 93.10, and 22.22% of convergent TCRs were antigen-specific, suggesting that convergent TCRs were much more likely to be antigen-specific than their non-convergent counterpart. We next estimated the statistical significance of this observation for each donor using Fisher's exact test (*Blevins and McDonald, 1985 Figure 2B*). The odds ratio for donor 1 reached 50.16, indicating that convergent TCRs are much more likely to be antigen-specific. Similar conclusions also held for donors 2 and 3, while the p-value (0.105) of donor 4 was less significant. Considering the small percentage of antigen-specific T cells detected in donor 4, it is possible that most antigens specific to donor 4's T cells did not fall within the spectrum of tested antigens in this experiment, which caused the lower overlap. Overall, the findings from this dataset supported the hypothesis that convergent T cells are more likely to be antigen-specific than other T cells.

We further tested this hypothesis using high-throughput bulk TCRβ-seq data. TCR-seq data were downloaded from ImmuneAccess, containing TCRβ sequences from over 1400 subjects that had been exposed to or infected with the SARS-CoV-2 virus, along with over 160,000 SARS-CoV-2-specific TCRs (*Snyder et al., 2020*). Similarly, we calculated the odds ratio of convergent TCRs and SARS-CoV-2-specific TCRs using Fisher's exact test for each sample. In comparison, we also investigated the overlap between clonally expanded T cells and antigen-specific T cells in these datasets, as T cell clonal expansion is a commonly used criterion for antigen specificity in recent clinical studies (*Kidman et al., 2020*; *Subudhi et al., 2016*). In each sample, TCR sequences with the highest read numbers were selected as expanded TCRs, and their number was restricted to the same number as convergent TCRs. Overall, the odds ratio of convergent TCR was almost unanimously higher than that of the clonally expanded T cells (*Figure 2C*). The average odds ratio of TCR convergence was 8.94, significantly higher than TCR clonal expansion, which was only 1.74 (*Figure 2D*). These results confirmed that converged TCRs are more likely to be antigen-experienced, and indicated that TCR convergence is a better indicator of antigen specificity than clonal expansion.

## Convergent T cells exhibit a CD8[+] cytotoxic gene signature

In the next step, we studied the gene expression signatures of the convergent T cells using single-cell RNA sequencing (scRNA-seq) data in the tumor samples to test the hypothesis that convergence is associated with an antigen-experienced effector phenotype. In the tumor microenvironment, antigen-specific T cells typically express cytotoxicity, memory, and/or exhaustion gene signatures (*Caushi et al., 2021*; *Oliveira et al., 2021*). We, therefore, checked the phenotypes of convergent T cells using recent scRNA-seq data generated from cancer studies.

The first dataset we investigated contained tetramer-labeled neoantigen-specific T cells in the MC38 tumor mouse model treated with neoantigen vaccine and anti-PD-L1 (*Liu et al., 2022*). The CD4[+] and CD8[+] T cells were analyzed separately using the same cell cluster annotations as previously described (*Liu et al., 2022*). We then defined and analyzed the convergent TCRs using the scTCR-seq data of the same T cells. T cells collected at tumor sites exhibited a much higher level of convergence than T cells in lymph nodes and spleen. Therefore, in the following description, we specifically refer to tumoral T cells. Only 1.85% (n=90) of the 4878 passed-QC CD4[+] cells had convergent TCRs (*Figure 3—figure supplement 1A*), which were not enriched in any cluster. In contrast, 9143 CD8[+] T cells passed the QC and 15.71% (n=1436) of them were convergent T cells (*Figure 3A*). Following this

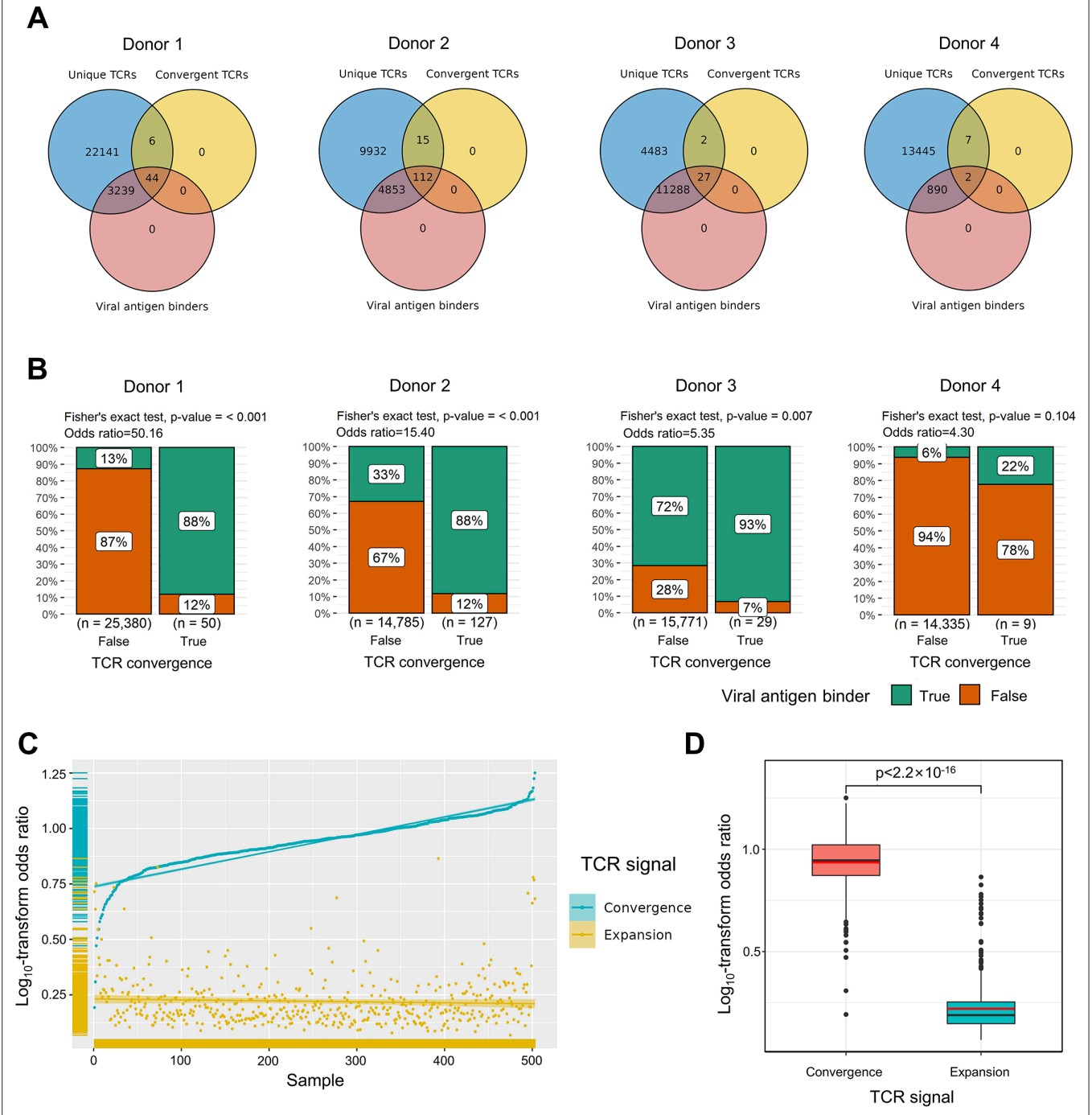

**Figure 2.** Convergent T cell receptors (TCRs) are more likely to be antigen-specific. (**A**) The overlaps between all unique TCRs, viral antigen-specific TCRs, and convergent TCRs of four donors, respectively. (**B**) The results of Fisher's exact tests regarding the association of convergent TCRs and viral antigen-specific TCRs of four donors. (**C**) Log10-scaled odds ratio of TCR convergence and expansion with respect to antigen-specificity of each sample. The samples were sorted in order of TCR convergence odds ratio from small to large. The horizontal stripes on the left represent the density of samples, colored by the TCR signal. (**D**) The difference of log10-transformed odds ratio between TCR convergence and TCR expansion. The statistical significance was calculated by Welch's t-test.

result, we noticed two interesting clusters: effector T cells with significant enrichment of convergent CD8+ T cells and naïve T CD8+ cells with none of the T cells having a convergent TCR (*Figure 3A*). To further investigate these two clusters, we re-clustered the CD8+ cells with a higher resolution (0.3) and divided the T cells into five new clusters (*Figure 3B*). 87.5% (419 out of 479) of T cells from cluster 04

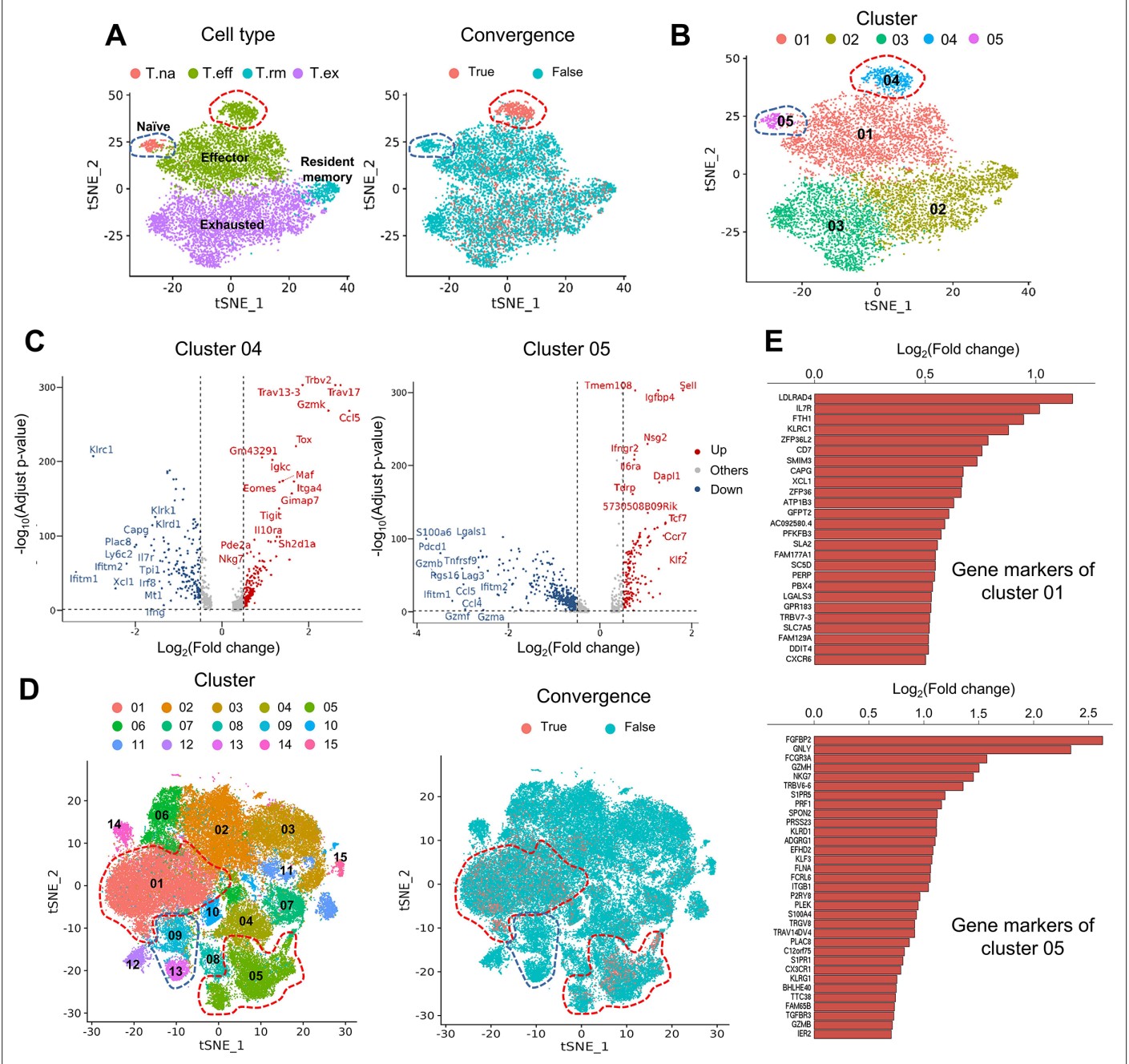

**Figure 3.** Convergent T cells exhibit a CD8[+] cytotoxic gene signature. (**A**) t-Distributed stochastic neighbor embedding (t-SNE) plots showing cell type annotations and convergent T cell distribution of CD8[+] T cells collected from mouse tumor tissues. (Each cell type corresponds to the following clusters in the original paper: T.na: CD8-04-Ccr7; T.eff: CD8-05-Fasl; T.rm: CD8-06-Xcl1; T.ex: CD8-08-Lag3; *Liu et al., 2022*) Each dot represents a T cell. The area within the red dashed line indicates enrichment of convergent T cells, and cells within the blue dashed line were naive T cells. (**B**) t-SNE displaying the re-cluster result of CD8[+] T cells. The red dashed line and blue dashed line highlighted the same groups of cells as in A. (**C**) Volcano plots showing the differentially expressed genes of cluster 04 and cluster 05. The p-value was calculated by the Wilcoxon test. (**D**) t-SNE plots showing the cell clusters and convergent T cell distribution of CD8[+] T cells collected from human tumors and normal adjacent tissues. An area within the red dashed line was enriched with convergent T cells, and the T cells within the blue dashed line were naive T cells. (**E**) The gene markers of cluster CD8-01-XCL1 and cluster CD8-05-FGFBP2 of human pan-cancer data. (Only the up-regulated genes of each cluster are shown).

The online version of this article includes the following figure supplement(s) for figure 3:

**Figure supplement 1.** T cell receptor convergence in CD4[+] T cells.

**Figure supplement 2.** The convergent T cell levels in CD4[+] and CD8[+] T cells.

**Table 1.** The Fisher's exact test result of CD8+ clusters.

| Clusters | Normal adjacent tissues | | | Tumor tissues | | |
|---|---|---|---|---|---|---|
| | Convergence ratio | Odds ratio | p-Value | Convergence ratio | Odds ratio | p-Value |
| CD8-01-XCL1 | 22.95% | 5.07 | <2.2e−16 | 15.03% | 5.66 | <2.2e−16 |
| CD8-02-GZMK | 2.59% | 0.24 | 1 | 3.48% | 0.85 | 0.9949 |
| CD8-03-TIGIT | 3.69% | 0.37 | 1 | 2.12% | 0.49 | 1 |
| CD8-04-ZNF683 | 7.43% | 0.79 | 0.9984 | 3.72% | 1.62 | <2.2e−16 |
| CD8-05-FGFBP2 | 32.44% | 6.19 | <2.2e−16 | 26.69% | 12.25 | <2.2e−16 |
| CD8-06-IFNG | 5.80% | 0.59 | 1 | 8.71% | 2.34 | 9.725E−15 |
| CD8-07-IHSPA6 | 22.75% | 3.93 | <2.2e−16 | 4.99% | 1.27 | 0.004411 |
| CD8-08-PRF1 | 4.84% | 0.5 | 1 | 3.11% | 0.77 | 0.9404 |
| CD8-09-IL7R | 1.02% | 0.1 | 1 | 1.44% | 0.35 | 1 |
| CD8-10-IFI6 | 11.11% | 1.23 | 0.2523 | 5.54% | 1.42 | 0.0003609 |
| CD8-11-STMN1 | 6.07% | 0.64 | 0.9603 | 3.93% | 0.98 | 0.5834 |
| CD8-12-KLRB1 | 0.48% | 0.03 | 1 | 0.13% | 0.03 | 1 |
| CD8-13-CCR7 | 0.00% | 0 | 1 | 0.00% | 0 | 1 |
| CD8-14-CXCR6 | 0.45% | 0.01 | 1 | 5.00% | 1.26 | 0.4325 |
| CD8-15-CTLA4 | 0.00% | 0 | 1 | 1.68% | 0.41 | 0.2 |

were convergent T cells, whereas none of the cluster 05 T cells (n=201) was convergent. Cluster 04 T cells expressed both effector and exhaustion markers, such as *Tox* (**Bordon, 2019**; **Sekine et al., 2020**), *Tigit* (**Ostroumov et al., 2021**), *Eomes* (**Li et al., 2018**), and inhibitory receptors like *Il10ra* (**Al-Abbasi et al., 2018**). On the other hand, cluster 05 mainly expressed naïve T cell gene signatures (**Al-Abbasi et al., 2018**), such as *Sell*, *Tcf7*, and *Ccr7* (**Figure 3C**), and down-regulated genes associated with effector function, like *Gzmb*, *Lgals1* (**Li et al., 2020**), and T cell activation or exhaustion (**Saleh et al., 2020**), like *Pdcd1* and *Lag3* (**Figure 3C**). Finally, by comparing the TCRs of cluster 04, 05 T cells and tetramer-sorted T cells specific to tumor neoantigen *Adpgk*, we observed a significant enrichment of convergent TCRs for neoantigen specificity in cluster 04 (OR = 4.41, p<0.001), while no enrichment in cluster 05 (**Figure 3—figure supplement 1B**). Based on these results, we concluded that convergent T cells are enriched for a neoantigen-activated effector phenotype in the tumor microenvironment.

To verify this observation in humans, we analyzed the second dataset of pan-cancer scRNA-seq data of nine cancer types (**Zheng et al., 2021**). There were samples collected from tumor tissues, normal adjacent tissues, and peripheral blood from human patients. 14 CD4+ clusters and 15 CD8+ clusters were defined using the Seurat R package, and the cluster annotations were assigned by the gene markers of each cluster shown in **Supplementary file 2**. Similar to the mouse data, convergent T cells were rare in CD4+ T cells (n=52,643), accounting for 0.22% (n=118) of the CD4+ population (**Figure 3—figure supplement 1C**), whereas 10.15% (n=7,065) of the CD8+ T cells (n=69,618) were convergent (**Figure 3D**). Convergent T cells were enriched in clusters CD8-01-XCL1 and CD8-05-FGFBP2, yet remained at low levels in the naïve clusters, CD8-09-IL7R and CD8-13-CCR7 (1.27 and 0.00%) (**Table 1**, the odds ratio and p-value were calculated by Fisher exact's test). CD8-01-XCL1 cluster expressed gene signatures of activated T cells, including *Il7r* (**Seddiki et al., 2006**), *ZFP36L2* (**Petkau et al., 2021**), *XCL1* (**Ordway et al., 2007**), and *CXCR6* (**Wang et al., 2021**; **Figure 3E**), as well as proliferation and mobility markers *LDLRAD4* (**Liu et al., 2017**) and *CAPG* (**Wei et al., 2020**). CD8-05-FGFBP2 cluster exhibited a CD8+ effector phenotype, upregulating genes involved in cytotoxicity and anti-tumor activity: *FGFBP2*, *GNLY*, *FCGR3A* (**Zheng et al., 2017**), *GZMH*, *PRF1*, *TGFBR3*, and *GZMB* (**Figure 3E**). In conclusion, the phenotypes of convergent T cells in human samples resembled those discovered in the mouse model. Convergent T cells were consistently enriched in the activated CD8+ T cell clusters with the gene signatures of cytotoxicity, memory, and exhaustion which demonstrated the phenotypes of antigen-specific T cells.

To validate the conclusion that TCR convergence was more prevalent in CD8[+] T cells than in CD4[+] T cells, we used another bulk TCRβ data collected from patients with classical Hodgkin lymphomas (cHLs) (*Cader et al., 2020*). As a result, CD8[+] T cells consistently exhibited a higher level of TCR convergence throughout different collection time points (*Figure 3—figure supplement 2A–2B*). This may be explained by larger cell expansions of CD8[+] T cells than CD4[+] T cells. Therefore, we calculated the number of convergent clones within CD8[+] T cells and CD4[+] T cells from the above datasets to exclude the effects of cell expansion. As a result, in the scRNA-seq mouse data, while only 1.54% of the CD4[+] clones were convergent, 3.76% of the CD8[+] clones showed convergence. Likewise, 0.17% of convergent CD4[+] T cell clones and 1.03% of convergent CD8[+] T cell clones were found in human scRNA-seq data. In the bulk TCR-seq cHLs data, similar results were also observed, where the gap between the convergent levels of CD4[+] and CD8[+] T cells narrowed but remained significant (*Figure 3—figure supplement 2C–2D*). In conclusion, these results suggest that CD8[+] T cells show higher levels of convergence than CD4[+] T cells, which substantiated our hypothesis that convergent T cells are more likely antigen-experienced. This observation has been tested using multiple datasets with diverse sequencing platforms and sequencing depth to minimize the impact of batch or other technical artifacts.

## TCR convergence is associated with the clinical outcome of ICB treatment

ICB has seen great success in treating late-stage cancer patients (*Bagchi et al., 2021*; *Jenkins et al., 2018*). While ICB treatment is effective for some patients, a significant proportion of patients remain unresponsive, and the reason for this is not completely understood. The discovery of a biomarker that can assist in the prognosis of immunotherapy remains one of the top clinical priorities. Since antigen-specific T cells play a crucial role in fighting tumor cells, we next investigated whether TCR convergence is predictive of the immunotherapy outcomes. We used bulk TCRβ-seq data generated by *Snyder et al., 2017*, which included PBMC samples from 29 urothelial cancer patients collected on the first day of the anti-PD1 treatment. We observed a significant association between TCR convergence level and both the overall survival (OS) (p=0.02) and progression-free survival (PFS) (p=0.00038) (*Figure 4A-B*). In the low TCR convergence group, over 90% of the patients suffered from disease progression within 75 days after initial treatment, whereas only 27% of the patients in the high TCR convergence group experienced disease progression at this time point (*Figure 4B*).

To validate the result, we examined another independent melanoma dataset containing 30 samples collected from melanoma patients treated with sequential ICB reagents (*Yusko et al., 2019*). Patients were randomly divided into two arms with the flipped ordering of anti-PD1 or anti-CTLA4 treatments (*Yusko et al., 2019*). TCR convergence significantly predicted patient outcome (p<0.0001) (*Figure 4C*), with higher TCR convergence associated with longer survival. Responders of the ICB treatment have significantly higher levels of TCR convergence (*Figure 4D*). To determine whether this link is confounded by other signals, we also included four additional variables that might influence the clinical outcomes in the multivariate Cox regression model, including TCR clonality, TCR diversity, different sequential treatments, and sequencing depth. TCR convergence remained significant (p=0.011) while adjusted for the additional variables (*Figure 4E*). Similarly, Cox regression models were applied to the urothelial cancer dataset above to adjust for the effects of TCR clonality, TCR diversity, as well as sequencing depth. As a result, TCR convergence remained a significant predictor of OS (p=0.045) and PFS (p=0.002) (*Figure 4—figure supplement 1A–1B*). Together, our results indicated that TCR convergence is an independent prognostic predictor for patients receiving ICB treatments.

## Discussion

Functional redundancy in crucial molecular pathways, such as cell cycle, metabolism, apoptosis, etc., is one of the key mechanisms that keeps biological systems robust and stable against genetic variations and environmental changes (*Duncan et al., 1999*; *Chambon, 1994*). Redundancy may occur at both molecular and cellular levels, the latter being particularly exploited by the immune system (*Dombrowski and Wright, 1992*). In this study, we investigated the convergence of TCRs, which could be a potentially unreported mechanism of redundancy in the adaptive immunity. The selection

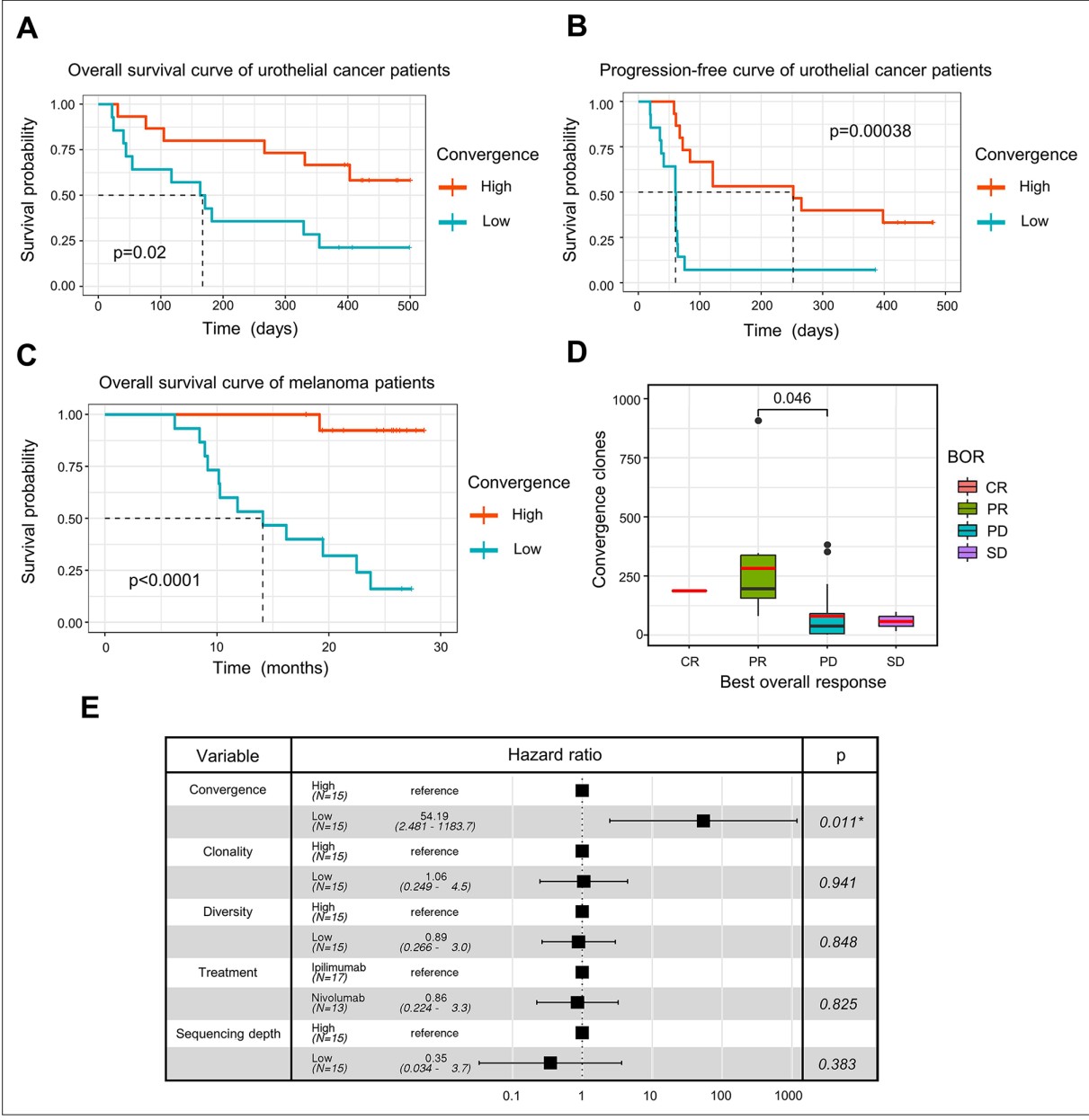

**Figure 4.** T cell receptor (TCR) convergence is associated with the clinical outcome of immune checkpoint blockade treatment. (**A**) Kaplan-Meier overall survival curves of urothelial cancer patients (n=29) with different levels of convergent T cells. The convergence levels were defined by the number of convergent T cell clones within each sample. High, above median; low, below the median. For TCR convergence, all the survival analyses in this study followed this definition. The p-values of all the Kaplan-Meier curves in this study were calculated by log-rank test. (**B**) Progression-free survival curve of urothelial cancer patients (n=29) with different levels of convergent T cells. The progression of the tumor was determined by mRECIST 1.1 test. (**C**) Overall survival curves of melanoma patients (n=30) with different levels of convergent T cells. (**D**) The differences in the TCR convergence levels among patients with the distinct best overall response toward anti-PD1 immunotherapy. The number of patients with each best overall response (BOR) is as follows: complete response = CR (n=1), partial response = PR (n=9), progressive disease = PD (n=18), stable disease = SD (n=2). The statistical significance was calculated by Welch's t-test. (**E**) Cox model multivariate analysis for the overall survival of melanomas cohort with covariate adjustments for other four variables: TCR clonality, TCR diversity, treatment method, and sequencing depth. The sequencing depth was determined by the total number of detected TCR clonotypes.

The online version of this article includes the following figure supplement(s) for figure 4:

**Figure supplement 1.** Multivariate cox regression analysis for urothelial cancer cohort.

of multiple clones of T cells with identical TCRs during antigen encounters might be a fail-safe mechanism to ensure the expansion of the antigen-specific T cells. This speculation is in line with our observations in this work: convergent TCRs are enriched for antigen-experienced receptors, and these T cells exhibited a neoantigen-specific, cytotoxic CD8$^+$ phenotype in the tumor microenvironment.

There are conceptual overlaps between TCR publicity and convergence, as both describe the redundancy of TCRs. However, our data suggested that these two processes could be fundamentally different. Currently, the functional consequence of V(D)J recombination bias remains unclear, but it has been speculated that this conserved process is evolved to eliminate common foreign pathogens, such as viruses and bacteria, that are frequently encountered in the environment (*Huisman et al., 2022*; *Li et al., 2012*). This process is genetically encoded and arises intrinsically. In contrast, TCR convergence might reflect the magnitude of clonal selection, which is highly contingent on the antigen landscape. Our observation that convergent and public TCRs diverge in cancers with high neoantigen load (*Figure 1D*) strongly supports this speculation.

Previous studies have demonstrated that CD8$^+$ and CD4$^+$ T cells may possess distinct TCRβ repertoire (*Wang et al., 2010*; *Emerson et al., 2013*), which results in the difference in their capacity to generate high avidity T cells (*Nakatsugawa et al., 2016*). Based on our findings in this study, CD8$^+$ T cells have a higher level of convergence than CD4$^+$ T cells on both single-cell and cell clone levels. In general, CD8$^+$ T cells play a direct role in killing abnormal cells (*Dustin and Long, 2010*; *Halle et al., 2017*), whereas most activated CD4$^+$ T cells function as conventional helper T cells or T regulatory cells to facilitate and regulate the immune response (*Wan, 2010*; *London et al., 1998*). This may lead to a greater impact of antigen selection on CD8$^+$ T cells than on CD4$^+$ T cells and thus a higher convergence level in cytotoxic CD8$^+$ T cells.

This indication of antigen-driven selection makes TCR convergence an attractive biomarker for disease diagnosis and/or prognosis. Many previous clinical studies have attempted to use the TCR repertoire to monitor the response of immunotherapies (*Page et al., 2016*; *Postow et al., 2015b*; *Hopkins et al., 2018*; *Roh et al., 2017*). TCR diversity indexes, such as Shannon's entropy, richness, clonality, etc. are routinely used in these studies, yet a strong association with the patient outcome is rarely observed. This is potentially because these indexes only reflect the overall dynamics of the repertoire, which cannot specifically capture the antigen-specific response. As a new summary statistic of the immune repertoire, TCR convergence hinges on the (neo)antigen-specific T cell responses in late-stage tumors and achieved better prediction accuracy compared to the diversity indexes. In addition, the potential prognostic value of TCR convergence and TCR similarity-based clustering was tested in previous studies (*Looney et al., 2019*; *Pogorelyy et al., 2019*). Awaiting further clinical validation, we anticipate that TCR convergence will be a powerful new biomarker to predict ICB therapy responses.

There are also limitations of this study. First, as TCR convergence is identified by small DNA changes, it relies heavily on sequencing accuracy. Improper handling of sequencing errors may result in the overestimation of TCR convergence (*Looney et al., 2019*). Second, since convergent T cells constitute only a small proportion of the total population of T cells, the study of TCR convergence requires a large number of sequenced T cells from each sample. Therefore, an accurate and deep sequencing approach is required for the study of TCR convergence. Third, although the phenotypic signatures of the convergent T cells were confirmed using both mouse and human scRNA-seq/scTCR-seq data, additional neoantigen-specific single T cell datasets would be necessary to consolidate our conclusions. Finally, our observation that TCR convergence is a favored prognostic predictor for ICB therapy was based on only three cohorts, due to the lack of qualified datasets. Future clinical studies with TCR repertoire profiles will be required to confirm TCR convergence as a new biomarker.

Antigen-specific T cells are central to a T cell immune response and determine the efficacy of immunotherapy, but their identification is challenging. However, the detection of convergent T cells is an easier task that only requires the profiling of the immune repertoire. The insights regarding TCR convergence gained from this work might provide an alternative angle to study the dynamics of antigen-specific T cells and a better understanding of the TCR repertoire.

## Materials and methods

### Datasets preparation

All the scRNA-seq or TCR-seq datasets used were accessed from public resources. This study analyzed 3 single-cell immune profiling datasets and 11 bulk TCRβ sequencing datasets, which were subjected to different analyses based on the features of the datasets. All the TCR-seq samples were downloaded from the immuneACCESS online database (https://clients.adaptivebiotech.com/immuneaccess). The detailed information for all datasets is described in *Supplementary file 1*.

### Definition of TCR convergence

T cells with identical CDR3 AA sequences and variable genes, but different CDR3 nucleotide reads were defined as convergent T cells. Single-cell immune profiling data analysis included pairing the α-β chain CDR3 sequences and variable genes of each T cell to form a complete TCR sequence. As for the bulk TCR-seq data, the β chain CDR3 sequences and their variable genes were used to represent their TCR sequences.

### Calculation of public TCRs proportion among convergent TCRs in different cohorts

The TCR sequences shared by at least 5% (n=34) of different individuals within the 666 samples from the Emerson cohort (*Emerson et al., 2017*) were defined as the public TCRs. The following datasets were constructed for each group. From immunoSEQ hsTCRB-V4b Control Data (*Hamm, 2020*), we selected 50 healthy samples with the deepest sequencing depths. 38 samples collected from whom have recovered from COVID-19 for more than 6 weeks were selected from the Nolan cohort (*Nolan et al., 2020*). The kidney cancer cohort (n=19) is a combination of data from our internal database and samples from the Chow cohort (*Chow et al., 2020*). Ovarian cancer data was also sourced from our in-house data (n=45). The immune sequencing data for blood cancer (n=53), lung cancer (n=50), and melanoma (n=29) came from the Cader cohort (*Cader et al., 2020*), Reuben cohort (*Reuben et al., 2020*), and Riaz cohort (*Riaz et al., 2017*), respectively. By dividing convergence-publicity overlap by the number of all convergent TCRs, we calculated the proportion of public TCRs among convergent TCRs in each sample.

### Definition of TCR clonality and diversity

TCR clonality was defined as 1-Pielou's evenness and was calculated as:

$$TCR\,clonality = 1 + \frac{\sum_i^n p_i log_2(p_i)}{log_2(n)},$$

where $p_i$ is the proportional abundance of unique TCR clonotype I and $n$ is the total number of TCR clonotypes within a given sample. TCR diversity was calculated as $n/N$, where $n$ is the total number of TCR clonotypes and $N$ is the total reproductive sequence reads.

### Analysis of scRNA-seq data

The scRNA-seq data were analyzed primarily using R/4.1.1 and Seurat 4.0.6. Data were processed with the conventional scRNA-seq pipeline before the analysis of TCR convergence. In brief, all data first had to pass QC, where cells with a high ratio of mitochondria and ribosomes were excluded since they were likely to be dying cells. Cells with too many or too few RNA reads were also excluded from studies because they might be doublets (a single barcode identifies more than one cell) or background noise. Data scaling and normalization were then performed. We regressed out the cell-cycle associated genes in our scaled data. Principal component analysis and t-distributed stochastic neighbor embedding (t-SNE) algorithm were applied to reduce the dimensions and visualize the data. Based on the markers of the cells within a cluster, cells were grouped into an appropriate number of clusters and assigned T cell types. Combining the scTCR-seq data with the scRNA-seq data allowed us to study the distribution of convergent T cells and their phenotypes.

### Statistical analysis

The two-tailed exact binomial tests were performed with the built-in function of R/4.1.1 to estimate the statistical differences in AA composition between convergent TCRs and non-convergent TCRs.

Fisher's exact tests were employed to analyze the association between convergent TCRs and antigen-specific TCRs, as well as the relationship between clonally expanded TCRs and antigen-specific TCRs. Using ggstatsplot v.0.8.0, the Fisher's exact test results of the scTCR-seq data were visualized. Ggsignif v.0.6.3 was used to calculate the statistical difference between different groups in all boxplots using Welch's t-test. In the 'box' within every boxplot, the black liner refers to the median, and the red line indicates the mean value. All survival analyses were conducted using the R package survival v.3.2.13. Survival curves were derived using the Kaplan-Meier method and the p-value was calculated by log-rank test. Cox multivariate regressions were performed to assess the association between the patients' outcomes and other variables.

## Acknowledgements

This work is supported by NCI 1R01CA245318 (BL) and 1R01CA258524 (BL).

## Additional information

### Funding

| Funder | Grant reference number | Author |
| --- | --- | --- |
| National Cancer Institute | 1R01CA245318 | Bo Li |
| National Cancer Institute | 1R01CA258524 | Bo Li |

The funders had no role in study design, data collection and interpretation, or the decision to submit the work for publication.

### Author contributions

Mingyao Pan, Resources, Data curation, Formal analysis, Validation, Visualization, Methodology, Writing - original draft; Bo Li, Conceptualization, Supervision, Writing - review and editing

### Author ORCIDs

Mingyao Pan ID http://orcid.org/0000-0003-2912-9599
Bo Li ID http://orcid.org/0000-0002-8617-900X

### Decision letter and Author response

Decision letter https://doi.org/10.7554/eLife.81952.sa1
Author response https://doi.org/10.7554/eLife.81952.sa2

## Additional files

### Supplementary files
• MDAR checklist

• Supplementary file 1. Details of datasets used in this study. Public datasets used in this study are summarized in this table. A total of 3 single-cell immune profiling datasets and 11 bulk TCR-seq datasets are involved in this study. The table lists the original project information and accession numbers for each dataset.

• Supplementary file 2. The clusters annotation in human pan-cancer data. The T cell cluster characteristics were summarized based on the gene expression signature of each cluster. This study only included the samples sequenced in the original study (Zheng et al., 2021), additional datasets used in the original publication that were sourced from other cohorts were not included in this study.

### Data availability

All data used in this work are publicly available. The Python code related to TCR convergence calculation are available at: https://github.com/Mia-yao/TCR-convergence/tree/main (*Yao, 2022*, copy archived at swh:1:rev:74d1132c3c8276c011afbe6e704587ae970099f5). The convergent TCR sequences of each cohort are uploaded to Zenodo, with https://doi.org/10.5281/zenodo.6603757.

The following dataset was generated:

| Author(s) | Year | Dataset title | Dataset URL | Database and Identifier |
|---|---|---|---|---|
| Yao P | 2022 | TCR convergence is a indicator of antigen-specific T cell response in immunotherapies | https://doi.org/10.5281/zenodo.6603757 | Zenodo, 10.5281/zenodo.6603757 |

The following previously published datasets were used:

| Author(s) | Year | Dataset title | Dataset URL | Database and Identifier |
|---|---|---|---|---|
| Nolan S, Vignali M, Klinger M, Dines J, Kaplan I, Svejnoha E, Craft T, Boland K, Pesesky M, Gittelman RM, Snyder TM, Gooley CJ, Semprini S, Cerchione C, Mazza M, Delmonte OM, Dobbs K, Carreño-Tarragona G, Barrio S, Sambri V, Martinelli G, Goldman J, Heath JR, Notarangelo LD, Carlson JM, Martinez-Lopez J, Robins H | 2020 | A large-scale database of T-cell receptor beta (TCRβ) sequences and binding associations from natural and synthetic exposure to SARS-CoV-2 | https://doi.org/10.21417/ADPT2020COVID | immuneAccess, 10.21417/ADPT2020COVID |
| Liu L, Chen J, Zhang H, Ye J, Lu C, Fu Y, Li B | 2022 | Concurrent delivery of immune checkpoint blockade improves tumor microenvironment for vaccine-generated immunity | https://www.ncbi.nlm.nih.gov/geo/query/acc.cgi?acc=GSE178881 | NCBI Gene Expression Omnibus, GSE178881 |
| 10x Genomics | 2022 | A new way of exploring immunity: linking highly multiplexed antigen recognition to immune repertoire and phenotype | https://www.10xgenomics.com/resources/datasets/cd-8-plus-t-cells-of-healthy-donor-1-1-standard-3-0-2 | 10X, cd-8-plus-t-cells-of-healthy-donor-1-1-standard-3-0-2 |
| Zheng L, Qin S, Si W, Wang A, Xing B, Gao R, Ren X, Wang Li, Wu X, Zhang Ji, Wu N, Zhang N, Zheng H, Ouyang H, Chen K, Bu Z, Hu X, Ji J, Zhang Z | 2021 | Pan-cancer single-cell landscape of tumor-infiltrating T cells | https://www.ncbi.nlm.nih.gov/geo/query/acc.cgi?acc=GSE156728 | NCBI Gene Expression Omnibus, GSE156728 |
| Emerson R, DeWitt W, Vignali M, Gravley J, Hu J, Osborne E, Desmarais C, Klinger M, Carlson C, Hansen J, Rieder M, Robins H | 2017 | Immunosequencing identifies signatures of cytomegalovirus exposure history and HLA-mediated effects on the T-cell repertoire | https://doi.org/10.21417/B7001Z | ImmuneAccess, 10.21417/B7001Z |

*Continued on next page*

*Continued*

| Author(s) | Year | Dataset title | Dataset URL | Database and Identifier |
|---|---|---|---|---|
| Reuben A, Zhang J, Chiou S, Gittelman RM, Li J, Lee W, Fujimoto J, Behrens C, Liu X, Wang F, Quek K, Wang C, Kheradmand F, Chen R, Chow C, Lin H, Bernatchez C, Jalali A, Hu X, Wu C, Eterovic AK, Parra ER, Yusko E, Emerson R, Benzeno S, Vignali M, Wu X, Ye Y, Little LD, Gumbs C, Mao X, XSong X, Tippen S, Thornton RL, Cascone T, Snyder A, Wargo JA, Herbst R, Swisher S, Kadara H, Moran C, Kalhor N, Zhang J, Scheet P, Vaporciyan AA, Sepesi B, Gibbons DL, Robins H, Hwu P, Sharma P, Allison JP, Baladandayuthapani V, Lee JJ, Davis MM, Wistuba II, Futreal PA, Zhang J | 2019 | Comprehensive T cell repertoire characterization of non-small cell lung cancer | https://doi.org/10.21417/AR2019NC | ImmuneAccess, 10.21417/AR2019NC |
| Li B | 2020 | Bulk TCRβ-seq of patients with renal cancers, Bulk TCRβ-seq of patients with high-Grade ovarian cancer | https://doi.org/10.5281/zenodo.3894880 | Zenodo, 10.5281/zenodo.3894880 |
| Cader FZ | 2020 | A peripheral immune signature of responsiveness to PD1 blockade in patients with classical Hodgkin lymphoma | https://doi.org/10.21417/FZC2020NM | ImmuneAccess, 10.21417/FZC2020NM |
| Hamm DE | 2020 | immunoSEQ hsTCRB-V4b Control Data | https://doi.org/10.21417/ADPT2020V4CD | ImmuneAccess, 10.21417/ADPT2020V4CD |

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
