## [Editor Report]

In this valuable and important study, the authors use cancer immunology datasets to study and discover a new biomarker for immune checkpoint blockade response. Not only does this work have the potential to be clinically impactful, but it also provides a deeper understanding of basic biology that can be applied to many different disease settings, and is supported by solid evidence.

---

## [Decision Letter]

**Decision letter after peer review:**

Thank you for submitting your article "T cell receptor convergence is an indicator of antigen-specific T cell response in cancer immunotherapies" for consideration by *eLife*. Your article has been reviewed by 3 peer reviewers, and the evaluation has been overseen by a Reviewing Editor and Tadatsugu Taniguchi as the Senior Editor. The reviewers have opted to remain anonymous.

Essential revisions:

1) In Figure 1D, use true healthy controls rather than COVID-19 subjects (I.e. the CMV dataset from Adaptive) unless there is good and substantial reason not to

2) Make all suggested changes regarding clarity (including confirming cluster IDs), and better contextualize the findings in relation to prior work.

*Reviewer #1 (Recommendations for the authors):*

The analysis is straightforward, yet generated really interesting results. I would recommend minor revisions before accepting it.

I have several questions that need the authors to clarify or provide more explanation.

1. In Figure 1 —figure supplement 1A, why after reaching 17 amino acids, did the degeneracy remain constant? The authors should discuss this in the Discussion section.

2. The scRNA-seq-based analysis is interesting. However, in Figure 3B, the recluster of CD8 T cells is not that different from Figure 3A. However, the convergent cells are in a totally different cluster. Are there any mislabeled clusters?

3. Why do CD4 T cells have very few degenerative clones? The authors should discuss this in the Discussion section.

*Reviewer #2 (Recommendations for the authors):*

Given the authors' track record of predicting cancer vs non-cancer using TCR repertoire, it will be interesting to investigate if and how TCR convergence can be used as an early-stage cancer biomarker.

There are a few suggestions to improve the scientific rigor and accessibility of this work:

1. In figure 1D, the author used COVID-19 patients' samples to represent non-Cancer patients. However, the inflammatory responses between COVID-19 patients and healthy individuals are expectedly different. It will be necessary to have true healthy donors as control.

2. In figure 3D, the author used a human dataset derived from multiple previous study cohorts. Convergent T cells were also sparsely distributed in cluster 01 and cluster 05. Since the cited dataset was a meta-analysis, it might be possible that some of these T cells may exhibit a 'tighter' distribution in a single cohort. It will be worthwhile to explore if the non-cluster-04 convergent T cells are enriched for other T cell phenotypes.

3. This study will benefit from more human datasets with neoantigen-specific T cells, both to verify the antigen-specificity of the convergent TCRs and to find out the types of antigens (viral or tumor antigens) that convergent TCR can recognize.

4. In the analysis of ICB response prediction (Figure 4), it will be interesting to investigate if there are similar or shared convergent TCRs across the responders. If so, these TCRs could be more direct predictors of immunotherapy outcome, although, given the large diversity of the antigen repertoire, this commonality might not be seen.

5. The authors should distribute an open-source package or script for users to easily analyze convergent TCRs.

*Reviewer #3 (Recommendations for the authors):*

Figure 1 typo: "covergence".

I think this may be a typo: "As antigen-specific T cells play a crucial role in defending tumor cells".

The definition of public TCRs needs to be justified: "Each of these cohorts was mixed with the 666 samples from the Emerson cohort(37) to form new cohorts with sample sizes of 716, 685, 711, 719, 716, and 695. The TCR sequences shared by at least 5% of different individuals within a cohort were defined as the public TCRs of that cohort." Assuming the "cohort" at the end of this sentence is the merged cohort, this seems like it could create artifacts due to the sizes/depths of the different cohorts, and therefore their contribution to their respective "merged" cohorts.

"starting with cysteine and ending with phenylalanine" Note that there is a common allelic variant (TRBJ2-7*02) that ends with a V.

The zenodo dataset file is in.rar format. It would be nice if a more standard format like.zip or.tgz could be used, given that there doesn't seem to be a public compression utility that understands rar files.

When contrasting convergent TCRs with sequence clustering approaches, the authors state "Nevertheless, the fact that T cells within each TCR convergent cluster share the same TCR amino acid sequence guarantees perfect antigen-specificity". For single-chain data, this isn't the case, since the paired chains could be different. So similarity and identity at the amino acid level are not really all that different.

---

## [Author Response]

Essential revisions:1) In Figure 1D, use true healthy controls rather than COVID-19 subjects (I.e. the CMV dataset from Adaptive) unless there is good and substantial reason not to

As suggested by the reviewing editor, we have included a true healthy control (unpublished immunoSEQ hsTCRB-V4b Control Data from Adaptive) in the analysis of Figure 1D. Besides, COVID-19 subjects were changed to samples collected from patients who have recovered from COVID-19 for more than 6 weeks as another independent cohort of non-Cancer samples, which are expected to have similar immunologic conditions as healthy individuals. The results of these two cohorts matched our previous conclusions.

2) Make all suggested changes regarding clarity (including confirming cluster IDs), and better contextualize the findings in relation to prior work.

We have made all suggested changes regarding clarity, including the cluster IDs of Figure 3A and Figure 3 —figure supplement 1A. These two figures were intended to cite the previous cluster annotation of the original paper. The cluster IDs of these two figures were removed to avoid potential confusion caused by the following re-cluster analysis. The cell cluster names of each cluster from original paper were described in the figure legend of Figure 3A and Figure 3 —figure supplement 1A:

Figure 3A: “Each cell type corresponds to the following clusters in the original paper: T.na: CD8-04-Ccr7; T.eff: CD8-05-Fasl; T.rm: CD8-06-Xcl1; T.ex: CD8-08-Lag3.”

Figure 3 —figure supplement 1A: “Each cell type corresponds to the following clusters in the original paper: T.na: CD4-04-S1pr1; T.Treg: CD4-05-Ctla4; T.Th1: CD4-06-Bhlhe40; T.rTreg: CD4-07-Ccr5.”

Reviewer #1 (Recommendations for the authors):The analysis is straightforward, yet generated really interesting results. I would recommend minor revisions before accepting it.I have several questions that need the authors to clarify or provide more explanation.1. In Figure 1 —figure supplement 1A, why after reaching 17 amino acids, did the degeneracy remain constant? The authors should discuss this in the Discussion section.

We thank the reviewer for bringing up this observation. We believe this invariance is likely due to a lack of detection power: The number of TCR sequences rapidly decreases after length reaching 17 amino acids and beyond (see Author response image 1). Specifically, length 18 CDR3s are only 1/6 in number of those with length 16. Since most convergent TCRs have a minimum degeneracy of two, when the number of CDR3s is lower, it is less likely to observe TCRs with a higher degeneracy. Therefore, the degeneracy of longer CDR3s would appear as unchanged.

**Author response image 1. sa2fig1:** 

2. The scRNA-seq-based analysis is interesting. However, in Figure 3B, the recluster of CD8 T cells is not that different from Figure 3A. However, the convergent cells are in a totally different cluster. Are there any mislabeled clusters?

We thank the reviewer for pointing out this potential confusion. To clarify, Figure 3A was intended to cite the cluster annotation of the original paper (Liu et al., Nat Can, 2022). In Figure 3B, we performed a reclustering using the same cells and a higher resolution, which separated cluster 04 from the original effector CD8 T cells. To lift this confusion, we have relabeled clusters of Figure 3A as well as Figure 3 —figure supplement 1A using the annotations in the original paper and revised the figure legend as follows:

Figure 3A: “Each cell type corresponds to following clusters in the original paper: T.na: CD8-04-Ccr7; T.eff: CD8-05-Fasl; T.rm: CD8-06-Xcl1; T.ex: CD8-08-Lag3.”

Figure 3 —figure supplement 1A: “Each cell type corresponds to following clusters in the original paper: T.na: CD4-04-S1pr1; T.Treg: CD4-05-Ctla4; T.Th1: CD4-06-Bhlhe40; T.rTreg: CD4-07-Ccr5.”

3. Why do CD4 T cells have very few degenerative clones? The authors should discuss this in the Discussion section.

As suggested by the reviewer, we have discussed the possible causes of the unbalanced convergence level between CD8^+^ and CD4^+^ T cells in the third paragraph of the Discussion section, which is shown below:

“Previous studies have demonstrated that CD8^+^ and CD4^+^ T cells possess distinct TCRβ repertoire (Emerson et al., 2013; Wang et al., 2010), which results in the difference in their capacity to generate high avidity T cells (Nakatsugawa et al., 2016). Based on our findings in this study, CD8^+^ T cells have a higher level of convergence than CD4^+^ T cells in both single cell and cell clone levels. These results suggest that TCRs of CD8^+^ and CD4^+^ T cells, despite appearing to serve the same purpose of antigen recognition, are under diverse selection pressures. In general, CD8^+^ T cells play a direct role in killing abnormal cells (Dustin & Long, 2010; Halle et al., 2017), whereas most activated CD4^+^ T cells function as conventional helper T cells or T regulatory cells to facilitate and regulate the immune response (London et al., 1998; Wan, 2010). This may lead to a greater impact of antigen selection on CD8^+^ T cells than on CD4^+^ T cells and thus a higher convergence level in cytotoxic CD8^+^ T cells.”

Reviewer #2 (Recommendations for the authors):Given the authors' track record of predicting cancer vs non-cancer using TCR repertoire, it will be interesting to investigate if and how TCR convergence can be used as an early-stage cancer biomarker.There are a few suggestions to improve the scientific rigor and accessibility of this work:1. In figure 1D, the author used COVID-19 patients' samples to represent non-Cancer patients. However, the inflammatory responses between COVID-19 patients and healthy individuals are expectedly different. It will be necessary to have true healthy donors as control.

As suggested by the reviewer, we have included a true healthy control (immunoSEQ hsTCRB-V4b Control Data from Adaptive) in the analysis of Figure 1D. Besides, COVID-19 subjects were changed to samples collected from patients who have recovered from COVID-19 for more than 6 weeks. These patients have passed the acute phase of immune responses and served as another independent cohort of non-Cancer samples. Using both cohorts, the new results matched our previous conclusions:

2. In figure 3D, the author used a human dataset derived from multiple previous study cohorts. Convergent T cells were also sparsely distributed in cluster 01 and cluster 05. Since the cited dataset was a meta-analysis, it might be possible that some of these T cells may exhibit a 'tighter' distribution in a single cohort. It will be worthwhile to explore if the non-cluster-04 convergent T cells are enriched for other T cell phenotypes.

We did the analysis suggested by the reviewer by excluding the cluster 04 cells and re-cluster and the rest of the T cells to see if convergent T cells are enriched for other T cell phenotypes. The results are shown in Author response image 2. After excluding the T cells from cluster 04, most convergent cells were enriched in exhausted T cells and resident memory T cells (p-value was calculated by right-tailed binomial exact test with ‘true probability of success is greater than expected probability’ as alternative hypothesis). These results are consistent with our hypothesis that convergence is enriched for antigen-experienced T cells.

3. This study will benefit from more human datasets with neoantigen-specific T cells, both to verify the antigen-specificity of the convergent TCRs and to find out the types of antigens (viral or tumor antigens) that convergent TCR can recognize.

We thank the reviewer for this constructive suggestion. As the reviewer would understand, human single T cell datasets with known antigen-specificity are extremely rare. Fortunately, there is a recent publication on this topic by Steven Rosenberg group (Lowery 2022, Science). In the past month, we requested controlled access of this dataset and performed in-depth convergence analysis of all the samples. However, the sample quality of most samples was considerably worse compared to mouse data with inadequate sequencing depth and limited number of detected convergent TCRs. The TILs were harvested at metastatic sites instead of the original site of tumorigenesis. Further, the annotation of neoantigen-specific TCRs in the original paper was incomplete, with small number of sequences not covering the full range of neoantigen-specific TCRs. We calculated the number of convergent TCRs and neoantigen-specific TCRs provided for each biospecimen (see Author response table 1), and unfortunately, all the numbers were too small for follow up analysis. Future high quality human dataset will be needed to further test our hypothesis.

**Author response table 1. sa2table1:** The number of convergent TCRs and neoantigen-specific TCRs in each biospecimen.

Biospecimen	Total cells	Convergent T cells	Convergence TCRs	Neoantigen-specific TCRs
4261	1864	0	0	2
4283	1083	10	1	11
4298	12594	75	14	11
4317	3558	76	6	5
4322	1401	9	3	5
4323	2004	8	2	19
4324	2593	9	2	7
4325	1124	4	1	0
4342	6814	37	11	7
4385	2696	26	2	2
4393	497	0	0	8
4394	1999	12	4	4
4397	2032	0	0	2
4400	1533	2	1	10
4421	7308	45	12	7

4. In the analysis of ICB response prediction (Figure 4), it will be interesting to investigate if there are similar or shared convergent TCRs across the responders. If so, these TCRs could be more direct predictors of immunotherapy outcome, although, given the large diversity of the antigen repertoire, this commonality might not be seen.

We followed the reviewer’s suggestion and performed this investigation. Unfortunately, due to the sparsity of convergence and small sample size, we only found four convergent TCR sequences that were shared by at least two responders in the melanoma cohorts, and thus it is not feasible to investigate their associations with the outcome. We thank the reviewer for also foreseeing this potential challenge and acknowledging the difficulties of this analysis.

5. The authors should distribute an open-source package or script for users to easily analyze convergent TCRs.

We have uploaded the python script to GitHub that every user can easily adapt to find convergent TCRs using the standard 10X output file ‘filtered_contig_annotations.csv’ (for scTCR-seq data) or standard.tsv file downloaded from ImmuneAccess (for bulk-TCR data) as input files. (https://github.com/Mia-yao/TCR-convergence/tree/main.)

Reviewer #3 (Recommendations for the authors):Figure 1 typo: "covergence".

We corrected this typo in Figure 1A.

I think this may be a typo: "As antigen-specific T cells play a crucial role in defending tumor cells".

We have corrected this typo as follows:

“Since antigen-specific T cells play a crucial role in fighting tumor cells”

The definition of public TCRs needs to be justified: "Each of these cohorts was mixed with the 666 samples from the Emerson cohort(37) to form new cohorts with sample sizes of 716, 685, 711, 719, 716, and 695. The TCR sequences shared by at least 5% of different individuals within a cohort were defined as the public TCRs of that cohort." Assuming the "cohort" at the end of this sentence is the merged cohort, this seems like it could create artifacts due to the sizes/depths of the different cohorts, and therefore their contribution to their respective "merged" cohorts.

We thank the reviewer for this very insightful suggestion. It is important to control the potential batch effects caused by different Adaptive data cohorts and sequencing depth. Therefore, we have redefined the public TCRs in this revision: instead of combining the Emerson cohort with each cohort and defining the public TCRs using the mixed cohort, we directly used the 666 samples from the Emerson cohort to define the same set of public TCRs for all the cohorts. We reanalyzed the public/convergence overlap and our conclusions remain unchanged.

"starting with cysteine and ending with phenylalanine" Note that there is a common allelic variant (TRBJ2-7*02) that ends with a V.

We agree that some allelic variants do not fall into our criteria of starting with a ‘C’ and ending with an ‘F’. To explore the influence of this allele, we calculated the percentage of TCRs passing this standard within desired CDR3 length. More than 99.4% of the total detected TCRs passed this standard (Author response image 3). This result suggests that CDR3s ending with V constitute a very small percentage of passed-QC TCRs profiled on the Adaptive Biotech platform. Therefore, the previous results are expected to be able to represent the overall population of TCRs.

**Author response image 3. sa2fig3:** 

The zenodo dataset file is in.rar format. It would be nice if a more standard format like.zip or.tgz could be used, given that there doesn't seem to be a public compression utility that understands rar files.

We have updated a new file of.zip format to zenodo with a new DOI number：

10.5281/zenodo.7139024.

When contrasting convergent TCRs with sequence clustering approaches, the authors state "Nevertheless, the fact that T cells within each TCR convergent cluster share the same TCR amino acid sequence guarantees perfect antigen-specificity". For single-chain data, this isn't the case, since the paired chains could be different. So similarity and identity at the amino acid level are not really all that different.

We agree with the reviewer and rephrased the aforementioned sentence to reflect the limitation of unpaired single chain TCR data:

“Nevertheless, the fact that T cells within each convergent cluster share the same CDR3β amino acid sequence likely represents heavily-shared antigen-specificity (Dash et al., 2017).”